# Transformer Fault Warning Based on Spectral Clustering and Decision Tree

**Hongli Liu [1], Junchao Chen [2]** **, Ji Li [1],\*, Lei Shao [1],\*, Lei Ren [1] and Lihua Zhu [1]**

[1]  Tianjin Key Laboratory for Control Theory & Application in Complicated Systems, Tianjin 300384, China
[2]  School of Electrical Engineering and Automation, Tianjin University of Technology, Tianjin 300384, China
\*  Correspondence: liji0606@tjut.edu.cn (J.L.); lei.shao@tjut.edu.cn (L.S.); Tel.: +86-22-13512883786 (J.L.);
   +86-22-18649178219 (L.S.)

**Abstract:** The insufficient amount of sample data and the uneven distribution of the collected data across faults are key factors limiting the application of machine learning in power transformer fault warning, as demonstrated by the poor adaptability of the established data-driven models under actual operating conditions. In this paper, an unsupervised and supervised learning method is designed for power transformer fault early warning based on electrical quantities and vibration signals. The method is based on the Fourier levels of transformer vibration signals under different electrical conditions measured in the field, and the vibration features are clustered according to their intrinsic properties by means of a spectral clustering algorithm. A decision tree model of the vibration characteristics under each cluster is then constructed to calculate early warning values for the transformer vibration spectrum under different electrical conditions, enabling the assessment of transformer production variability. The above process, which is based on field measurement data and data mining analysis methods, is cheaper than the existing transformer fault warning techniques at home and abroad and makes better use of information and training models.

**Keywords:** transformer vibration; spectral clustering algorithm; decision trees; vibration characteristics; fault warning

## 1. Introduction

Fault-warning technology based on the online monitoring of vibration signals can find early faults in transformers in a timely and effective way. This is an important function in substations and power inspections, and it is also very important for preventing sudden and large power outages caused by poor operating conditions of transformers, which can even lead to accidents such as explosions and fires [1–5].

Fault warning tasks usually consist of two phases: offline modeling and online monitoring. In the offline modeling phase, a transformer operating mechanism model is trained using a section of normal data to determine the boundaries of the normal data; for online monitoring, if the data to be detected exceeds the established boundaries, there is a high probability that a fault has occurred in the transformer in that operating condition [6]. With the concept of the ubiquitous power IoT, various sensing and monitoring technologies have been rapidly developed, and the transformer vibration signal data have gradually shown big data characteristics such as large volume, high dimensionality, and fast growth, while traditional diagnosis technology has problems of low efficiency and high cost. Therefore, data mining and analysis based on actual operating conditions and the construction of low-cost, generalized transformer condition identification and fault early warning models are of great importance to ensure the stable operation of power systems and a quality power supply. The laboratory-based vibration test platform is a common method used by scholars at home and abroad to study transformer fault early warning [7]. In [8], a 110 kV power transformer that was producing GDR II warnings underwent a number of electrical and chemical tests in order to be examined for defects. In [9], for the modeling

and real-time application of fault diagnosis within the transformer, a single-phase, 600 VA, 220/110 V, 50 Hz transformer was employed in the lab. However, in actual engineering, transformer state change is often the result of an accumulation of weak faults; state change is a gradual process, and with many transformer models and complex structures, most studies can only be carried out under limited conditions to simulate the special vibration situations of transformers, so data-driven models trained based on laboratory-established fault samples are not sufficiently adaptable to perform at some sites.

On the other hand, international researchers in the field of power transformer defect warning have focused a lot on data-driven models, such as artificial neural networks, support vector machines, random forests, and principal component analysis, to solve these problems better [10–13]. In [14], a training technique for deriving rules from a functionally approximated ANN utilizing the concentration of dissolved gases in transformer oil as the input is suggested in order to implement fault warning and defect diagnostics in transformers using artificial neural networks. However, the synchronization of the model parameters can be difficult to control and operates slowly for the neural network approach; in contrast, the defect warning strategy based on the SVM algorithm operates quickly and accurately. In [15], in the SVM-BA optimized SVM model for oil-immersed transformers, the kernel function and penalty margin are integrated with the Gaussian classifier. However, supervised learning algorithms such as these have a strong reliance on the completeness of sample information, and monitoring data in real industrial settings often lacks appropriate data labels. Unsupervised learning and semi-supervised learning can be applied to condition monitoring data where fault data are scarce or lacking in labels [16]. In [17], to handle the challenging cases that are largely unclassifiable by Duval's triangles, a novel DGA diagnostic method based on the K-means method with an enhanced KNN cumulative voting mechanism was created. This method is appropriate for the early warning of transformer defects. In [18], to analyze and process transformer DGA data, the upgraded FCM algorithm is employed, significantly resolving the issue of classic clustering methods not performing as expected in tasks requiring transformer defect warning and diagnosis.

The number of faulty samples is very scarce, which in turn leads to an uneven distribution of the actual equipment data set. When machine learning methods are applied to such unbalanced samples, the training model is biased towards the majority class and does not perform well for the minority class. Therefore, a good number of samples is a necessary condition for ensuring that the above machine learning algorithms produce models that work well in real life.

Based on the above analysis, the data mining concept is used in this paper to study the vibration characteristic values of transformers under different operating conditions based on the field measurement data of transformer vibration signals and the existing transformer vibration mechanism model of power quality. Firstly, a spectral clustering algorithm is used to cluster the transformer vibration signal data set to achieve the division of the field-measured transformer vibration signal into working conditions. A decision tree model is then constructed to analyze the vibration characteristics of the transformer during harmonic current, light load, heavy load, and three-phase unbalanced current operation. The method establishes a direct link between the transformer's operating state and each of its amplitude and frequency characteristic quantities. It also gives a way to keep track of the transformer's state and warn of problems using vibration signals during gradual state changes.

## 2. Model Selection and the Basic Algorithm Flow

### 2.1. Transformer Vibration Mechanism Model Based on Electrical Quantities

To comprehensively and accurately analyze the vibration propagation characteristics of transformers, two types of electrical quantities, namely transformer voltage and current, need to be taken into account, and the vibration sources need to be located based on the different characteristic electrical quantities. Therefore, this paper chooses typical power

quality problems in the grid, such as voltage distortion, harmonic current, and three-phase unbalance, and rebuilds the vibration propagation mechanism model with different types of electrical quantities that can reflect the load variation of a light load and a heavy load. This shows the relationship between electrical quantities and vibration signals, as shown in Figure 1.

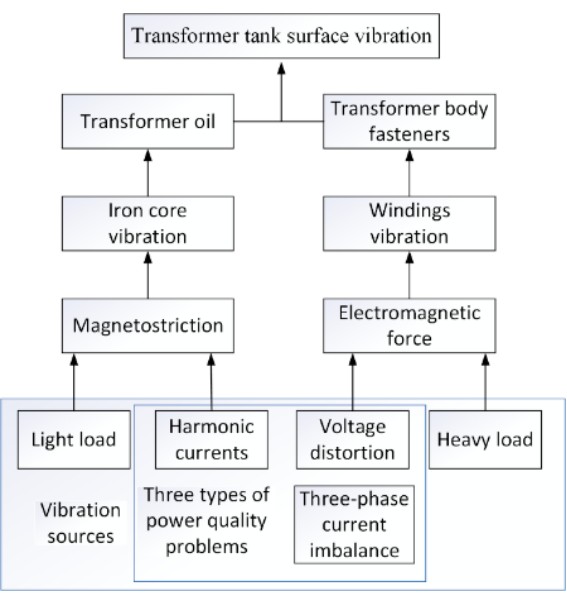

**Figure 1.** Transformer vibration mechanism model based on power quality.

### 2.2. Electrical Quantity-Based Transformer Vibration Fault Warning Process

The two stages of the algorithm development process are feature extraction and algorithm identification. The creation of a complete defect warning mechanism follows the construction of a transformer vibration feature model for each operating situation and the establishment of alarm thresholds. Figure 2 depicts the big picture.

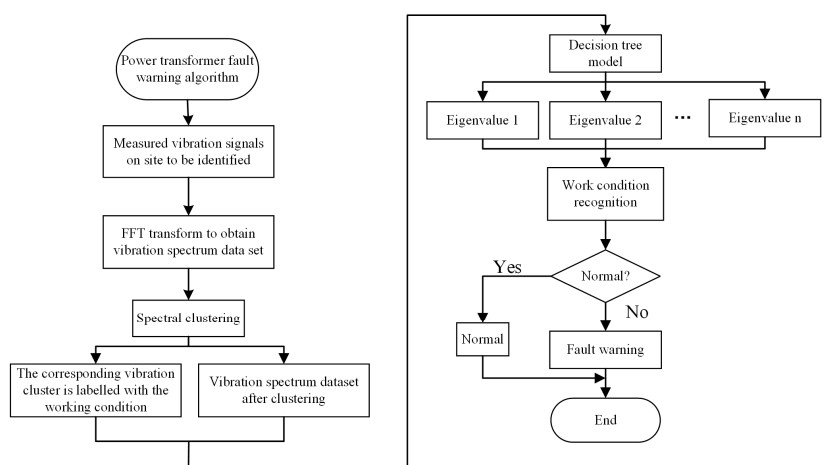

**Figure 2.** Flow chart of transformer fault early warning algorithm.

## 3. Clustering of Transformer Vibration Signal Feature Quantities

### 3.1. Principle of the Spectral Clustering Algorithm

Spectral clustering (SC) is one of the typical representatives of modern clustering algorithms based on graph theory, which inherits the traditional algorithm idea and combines it with graph theory optimization theory, greatly improving the universality of clustering

algorithms. In power systems, spectral clustering has been widely used in areas such as black start, load classification, and ultrashort-term wind speed prediction [19].

The essence of spectral clustering is to transform the clustering problem into a graph optimal partitioning problem, which is a method of cutting the graph based on matrix eigenvectors and according to the edge weights between vertices [20]. The basic idea of spectral clustering is to treat the original data set to be processed as an undirected weighted graph in space $G(V, E)$, which is the set of data points in the dataset $V = (V_1, V_2, \cdots, V_n)$, the set of vertices in the undirected weight graph, and the set of edges. Most spectral clustering algorithms use the fully connected (FC) method to construct the adjacency matrix $W$, where the element values in the matrix represent the degree of similarity between the data points. Use the Gaussian kernel function RBF in FC to define the edge weights between any two points:

$$W_{ij} = \begin{cases} \exp\left(-\frac{\|v_i - v_j\|_2^2}{2\sigma^2}\right), & i \neq j \\ 0, & i = j \end{cases} \tag{1}$$

where $v_i$, $v_j$ are any two sample points of the data set $V$, and $\sigma$ is a fixed scale parameter in the Gaussian kernel function.

The degree of a vertex is the sum of the connection weights of that vertex and other vertices, and using the definition of the degree of each data point, the degree matrix can be obtained, $D$. This gives the Laplacian matrix, also known as the Kirchhoff matrix, as a matrix representation of the graph:

$$L = D - W \tag{2}$$

Dividing the data set into problems by introducing indicator variables translates into solving the optimal indicator vector matrix $H$, solving the problem of difficult NPs in the optimal division of the atlas. The $k$-class normalized tangent diagram can be transformed into the following model:

$$\begin{cases} \min_{H \in \mathbb{R}^{m \times k}} Tr(H^T L H) \\ s.t\ H^T D H = I \end{cases} \tag{3}$$

where $Tr$ is the trace of the matrix and $H$ is the vector matrix. Model (3) is a standard problem for minimizing the trace. According to the Laplace matrix property, this optimization problem can be transformed into finding the minimum first $k$ eigenvalues of $D^{-1/2} L D^{-1/2}$ and normalizing the corresponding eigenvectors to obtain the final eigenmatrix $F$. The clustering result is obtained by performing K-means clustering on the eigenmatrix $F$.

### 3.2. Extraction of Amplitude–Frequency Characteristic Quantities of Transformer Vibration Signals

For experimental investigation, the vibration signals of a 10 kV three-phase wound transformer were recorded under four different operating conditions: harmonic current, light load, heavy load, and three-phase unbalance. Three sets of vibration signals were recorded for each condition. Each experiment used 40 sets of data with a 200 ms time window, each of which was divided into 60 s of recorded data at a 10 kHz sampling rate. In order to acquire the amplitudes of the harmonic components of transformer vibration from 50 Hz to 2400 Hz, with 50 Hz as the fundamental waveform, for a total of 48, a fast Fourier transform was carried out on each 200 ms window of vibration time domain data. The results are presented in Figure 3.

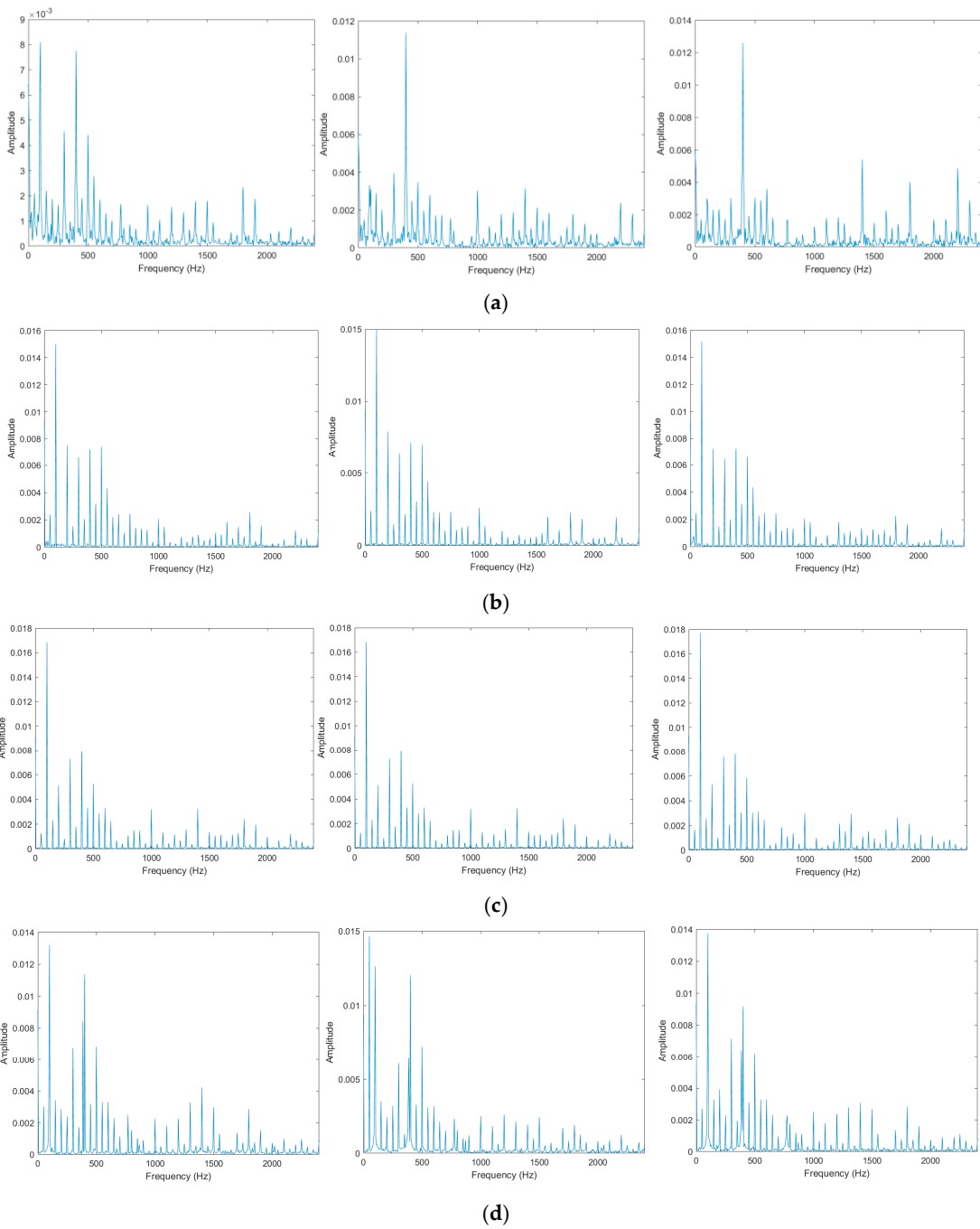

**Figure 3.** Frequency domain characteristic diagram of transformer vibration. (**a**) Spectrum during harmonic current operation. (**b**) Spectrum during light load operation. (**c**) Spectrum during heavy load operation. (**d**) Spectrum during three−phase unbalance operation.

The following matrix is produced by the Fourier transform using the time domain data for each set of 40 windows.

$$\overline{C}_{Fea} = [C_1, C_2, \cdots, C_n, \cdots, C_{40}] \tag{4}$$

In Equation (4), the matrix $\overline{C}_{Fea}$ represents different types of electrical quantities, such as harmonic currents, etc. The vector $C_n$ is the Fourier transform amplitude information for the $n$th vibration time domain data window, $n \in [1, 40]$, as shown in Equation (8) below.

$$C_n = [c_{n-1}, c_{n-2}, \cdots, c_{n-m}, \cdots c_{n-48}] \tag{5}$$

In Equation (5), $C_{n-m}$ is the amplitude of each vibration harmonic component corresponding to the amplitude–frequency curve in Figure 3, i.e., the $n$th vibration harmonic amplitude of the $m$th time window.

### 3.3. Extraction of Amplitude–Frequency Characteristic Quantities of Transformer Vibration Signals

The spectrum data of four different transformer vibration signals—harmonic, light load, heavy load, and three-phase unbalance—obtained following FFT processing, with a total of 440 groups, are pooled as the sample data set in this study after their operating condition labels have been removed. The shape of the distribution of the data samples in space is observed by means of a scatter plot. In unsupervised learning, the principal component analysis (PCA) algorithm is used to "downscale" the data set. This maps the high-dimensional data to the low-dimensional space, which allows one to see how the sample data set is distributed in the low-dimensional space.

The shape of the sample distribution shown in Figure 4 reflects the fact that the sample data set is nonconvex and irregularly distributed in space. Noisy data and local optimal solutions can make it difficult to find similarity measures and iterate them through Euclidean distances, Manhattan formulas, Jaccard coefficients, and so on when working with these kinds of data sets.

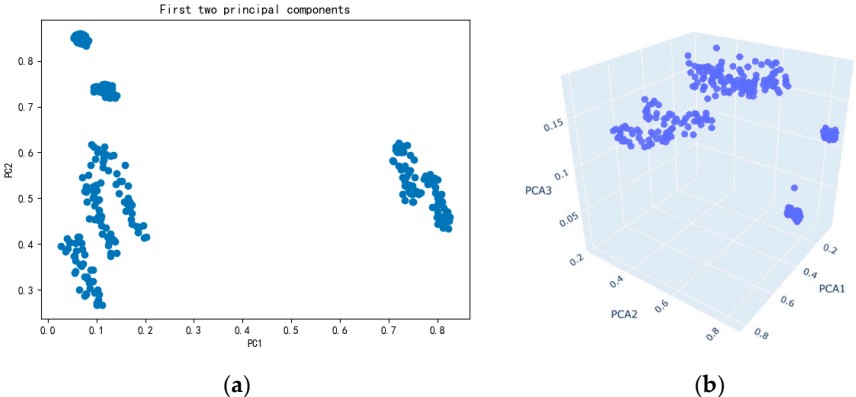

|       |       |
| :---: | :---: |
| (**a**) | (**b**) |

**Figure 4.** Scatter diagram of the vibration spectrum. (**a**) Two-dimensional scatter plot. (**b**) Three-dimensional scatter plot.

Two algorithms, K-means, and fuzzy clustering [21] are popular unsupervised learning algorithms in data mining analysis and are widely used in image processing, material exploration, and simulation prediction.

Three different data sets taken from transformer vibration signals at 50 Hz−400 Hz, 50 Hz−1250 Hz, and 50 Hz−2400 Hz were used to test the clustering effects of three algorithms: spectral clustering, K-means, and FCM.

As a point of comparison, a cluster count of 4 in the horizontal coordinate was used because the elbow approach revealed that the ideal number of clusters for the transformer vibration spectrum data set was 4. As evaluation markers of the clustering effect, the Davies–Bouldin index (DBI) and the silhouette coefficient were employed. The better the clustering effect, the closer the similar samples are to one another and the further apart the different samples are from one another; on the other hand, the better the clustering effect, the smaller the value of DBI, the closer the similar samples are to one another, and the farther apart the different samples are from one another.

Combining Figure 5 and Table 1, the following conclusions can be drawn.

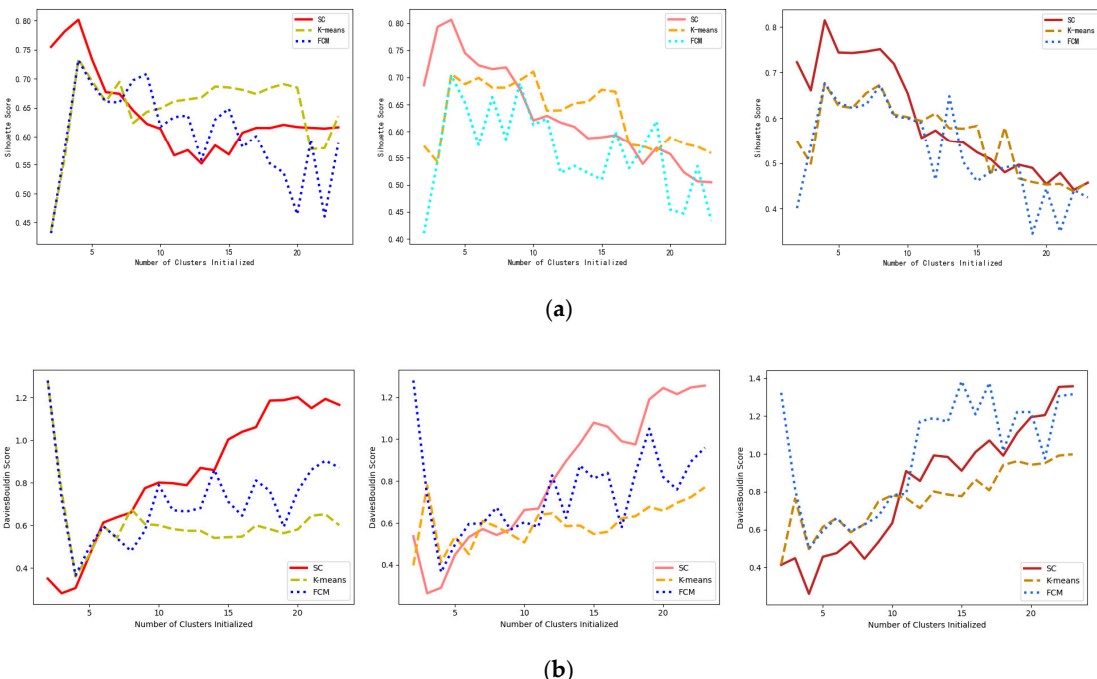

**Figure 5.** Contour coefficient map based on different clustering algorithms. (**a**) Plot of silhouette coefficients for different dimensional datasets. (**b**) Plot of Davies–Bouldin indexes for different dimensional datasets.

**Table 1.** Comparative metrics of clustering effects of different algorithms.

| Algorithm | Data Set Dimension | Silhouette Coefficient | Davies–Bouldin Index |
|---|---|---|---|
| Spectral clustering | 8 | 0.7870 | 0.4466 |
| K-means | 8 | 0.8100 | 0.2840 |
| FCM | 8 | 0.7241 | 0.3577 |
| Spectral clustering | 25 | 0.8342 | 0.3430 |
| K-means | 25 | 0.7791 | 0.3429 |
| FCM | 25 | 0.6718 | 0.3933 |
| Spectral clustering | 48 | 0.9267 | 0.1548 |
| K-means | 48 | 0.6876 | 0.3960 |
| FCM | 48 | 0.6765 | 0.4989 |

① The performance of the spectral clustering method eventually surpasses that of the K-means and FCM algorithms as data amount and dimensions rise.

② A comparison of the contour coefficients of the three clustering algorithms under different categories is given in Figure 5a. For all three algorithms, the maximum value of the silhouette coefficient is taken at a cluster class number of four. When there are few cluster classes, the values of the spectral clustering algorithms' silhouette coefficients are all higher than those of the K-means and FCM algorithms.

A comparison of the DBI of the three clustering algorithms for different classes is given in Figure 5b. The spectral clustering algorithm takes the least value of DBI at a cluster class number of four. When there are few cluster classes, the spectral clustering algorithm's total DBI values are lower than those of the K-means and FCM algorithms.

In summary, for the spectral dataset of transformer vibration signals, the spectral clustering algorithm can better measure the similarity between vibration spectrum data and has a better clustering effect and stability.

## 4. Decision Trees

As a top-down supervised learning classification algorithm, the inherent characteristics of the decision tree algorithm make it insensitive to the true or nonlinear characteristics of

the data, taking into account the interactions between variables while also providing a clear and intuitive representation of the relationship between logical labels and feature vectors in the form of a tree diagram, enhancing the mapping relationship between vibration features and transformer operating conditions. So, using decision trees to build a classification model for transformer operating conditions can help make the analysis of data after spectral clustering clearer and more accurate [22].

In this paper, a decision tree algorithm with information entropy and information gain as splitting rules is used [23]. The type of working condition is used as a category attribute, and other relevant factors, such as the amplitude of vibration harmonics components from 50 Hz to 2400 Hz, are used as non-category attributes to construct a transformer fault warning decision tree. The specific steps are as follows:

Step 1: Determine the sample set $D$. The collected transformer vibration harmonic component amplitudes from 50 Hz to 2400 Hz and the working condition category are composed into a complete sample so that a large amount of actual data can form the sample set.

Step 2: Calculate the sample information expectation. The sample information entropy is calculated using Equation (6).

$$Ent(D) = -\sum_{i=1}^{|D_n|} p(A_i) \log_2 p(A_i) \tag{6}$$

In Equation (6): $D$ is the sample data set; $|D_n|$ is the total number of transformer operating conditions types; $A_i$ is the amplitude of the harmonic component of the transformer vibration from 50 Hz to 2400 Hz for working condition type $i$; and $p(A_i)$ is the number of samples with condition type $i$ as a proportion of the total number of samples.

Step 3: Calculate the information gain for each non-category attribute. In this sample data set, there are 440 values of the non-categorical attribute $K$. According to the value of this attribute, the sample set of transformer condition categories can be divided into 440 parts, and the following formula is used to calculate the information gain of each non-categorical attribute.

$$Ent_K(D) = \sum_{l=1}^{440} \frac{|D_l|}{|D|} \times Ent(D) \tag{7}$$

$$Gain(D, A) = Ent(D) - Ent_K(D) \tag{8}$$

In Equation (8), where there are $N$ values of the eigenvalue $A$: $|D^N|$ is the $N$ branch containing all samples with value $A$ on $A^N$ in the sample data set $D$; and $\frac{|D^N|}{|D|}$ is the weight of the $N$ branch node.

Step 4: Select the split property node. For the non-class attributes, the amplitude of each vibrational harmonic component of the transformer from 50 Hz to 2400 Hz is calculated according to the method shown in Step 3, the sample information gain corresponding to each attribute is determined, and the non-class attribute with the greatest gain is selected as the split node.

## 5. Experimental Analysis

To ensure the homogeneity of the test samples, the spectral amplitudes of the samples after FFT and removal of the working condition labels and merging were normalized using a linear transformation.

$$X_{norm} = \frac{X - X_{\min}}{X_{\max} - X_{\min}} \tag{9}$$

where $X_{\max}$ and $X_{\min}$ are the maximum and minimum values of the spectral amplitude in the sample, respectively.

The vibration spectrum data set obtained after the normalization transformation was spectrally clustered to obtain four different types of vibration cluster classes. The results are shown in Table 2.

**Table 2.** Statistics of sample points in the cluster.

| Cluster Class Number | Number of Sample Points |
|:---:|:---:|
| Cluster 1 | 120 |
| Cluster 2 | 120 |
| Cluster 3 | 120 |
| Cluster 4 | 80 |

The experimental results show that spectral clustering divides the transformer vibration signal spectrum data set into four vibration clusters. Based on how the original data set was put together, the four clusters are called light load, heavy load, harmonic current, and three-phase current unbalance.

According to the transformer vibration signal clustering thermogram in Figure 6, the spectrum at odd frequencies such as 50 Hz, 150 Hz, 250 Hz, and 400 Hz is stronger under the unbalanced three-phase current condition, where the odd frequency component is caused by the zero-sequence current invasion during the asymmetric operation of the system. Under harmonic current conditions, the spectrum is higher at 100 Hz, 300 Hz, 400 Hz, and above 1000 Hz for high-frequency harmonic components, as the core magnetostriction generates 100 Hz, 200 Hz, and other vibrational harmonics, and harmonic currents often lead to superimposed harmonic components on the winding amperage. Under light load and heavy load conditions, the vibration signal is mostly made up of even frequency components between 100 Hz and 500 Hz. However, under heavy load conditions, the combination of winding vibration and core vibration caused by a current in the winding will cause a high frequency vibration component of 1000 Hz or more through nonlinear propagation.

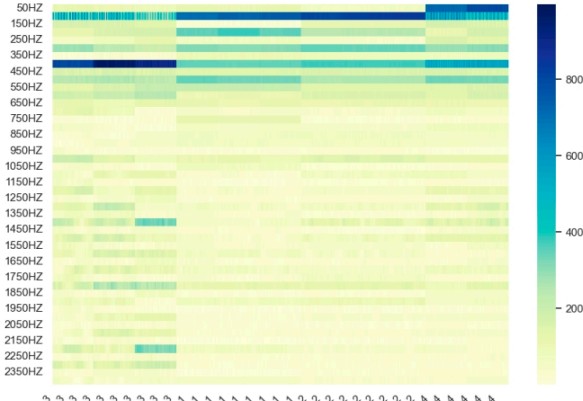

**Figure 6.** Clustering heat map of transformer vibration characteristics.

On the basis of the vibration clusters of each transformer operating condition, the vibration spectrum amplitude is converted into discrete variables consisting of "low", "medium", and "high" by using the trilateration method. The information gain $Gain(D, A_{400\ \text{Hz}}) = 0.977$ purity of the vibration amplitude at 400 Hz is calculated to be the highest among the 48 vibration features, so 400 Hz can be used as the root node of the transformer vibration feature decision tree model. The information gain at 50 Hz and 350 Hz in the second layer is 0.854 and 0.730, respectively, while the information gain at 50 Hz in the third layer is 0.222. The transformer vibration feature decision tree model is shown in Figure 7.

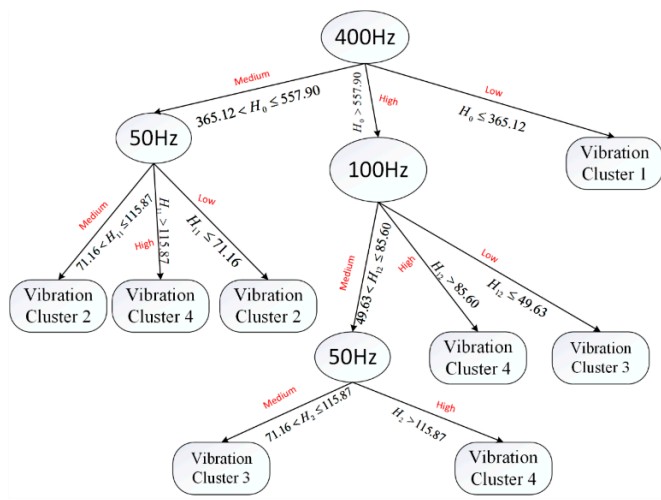

**Figure 7.** Decision tree of transformer vibration characteristics.

In the diagram, $H_0, H_{11}, H_{12}, H_2$ represent the amplitude of the vibration spectrum at 400 Hz, 50 Hz, 350 Hz, and 50 Hz, respectively. Among the four transformer operating conditions reflected in Figure 7—light load, heavy load, harmonic current, and three-phase current unbalance—the 400 Hz vibration signal amplitude in the light load condition is relatively low compared to the other three conditions and is therefore first split out at the root node in the decision tree. When the transformer operating condition satisfies $H_0 \leq 365.12$, the transformer is in the light load operating condition.

The three-phase unbalance is divided into high-spectral components at 50 Hz and 350 Hz in the decision tree. Combining the clustering heat map with the mechanism analysis, the spectral components of odd harmonics such as 50 Hz, 150 Hz, and 250 Hz can be used as the vibration characteristics of the transformer's three-phase current unbalance, i.e., the transformer is in a three-phase unbalanced operation when the transformer operation state meets $365.12 < H_0 \leq 557.90$ at $H_0 > 557.90$, or $H_0 > 557.90$ at $H_{12} > 85.60$, or $H_0 > 557.90$ at $49.63 < H_{12} \leq 85.60$ and $H_2 > 115.87$.

Although there is a certain 50 Hz vibration component in the heavy load condition, the difference between the 50 Hz vibration signal amplitude and that in the three-phase unbalance is large, so the heavy load and three-phase current unbalance conditions can be judged at the 50 Hz leaf node. When the transformer operating state meets $365.12 < H_0 \leq 557.90$ at $71.16 < H_{11} \leq 115.87$ or $H_{11} \leq 71.16$, the transformer is in the heavy load operating state.

This establishes the state classification of the four operating conditions of the transformer, as well as the safety thresholds for the 50 Hz, 350 Hz, and 400 Hz spectral components for each operating condition. If the range of the relevant state parameters in the decision tree model built from the collected vibration signals or the state identification nodes changes, a fault warning is sent out about the transformer's state.

The transformer vibration signal in two unknown states is acquired for characterization and combined with the transformer vibration mechanism analysis to diagnose the type of fault. In the fault state, the voltage, sound field, magnetic field, and other relevant parameters are measured. The transformer is then modeled and simulated with finite element simulation software to look at the magnetic field, coil force, and other properties in this state to see if the above conclusion is correct [24].

Figure 8a,b demonstrate that when the transformer is operated in state 1, the vibration signal in the original light load state has a significantly higher spectral component at frequencies between 100 and 500 Hz, whereas when the transformer is operated in state 2, the vibration signal only has a significantly higher spectral component at frequencies between 100 Hz and 200 Hz. The vibration signals produced by the transformer in these two unknown states result in a decrease in the variability of the vibration characteristics between

the various clusters in the vibration data set as compared to the vibration characteristics of the different operating conditions in the transformer's normal operating state.

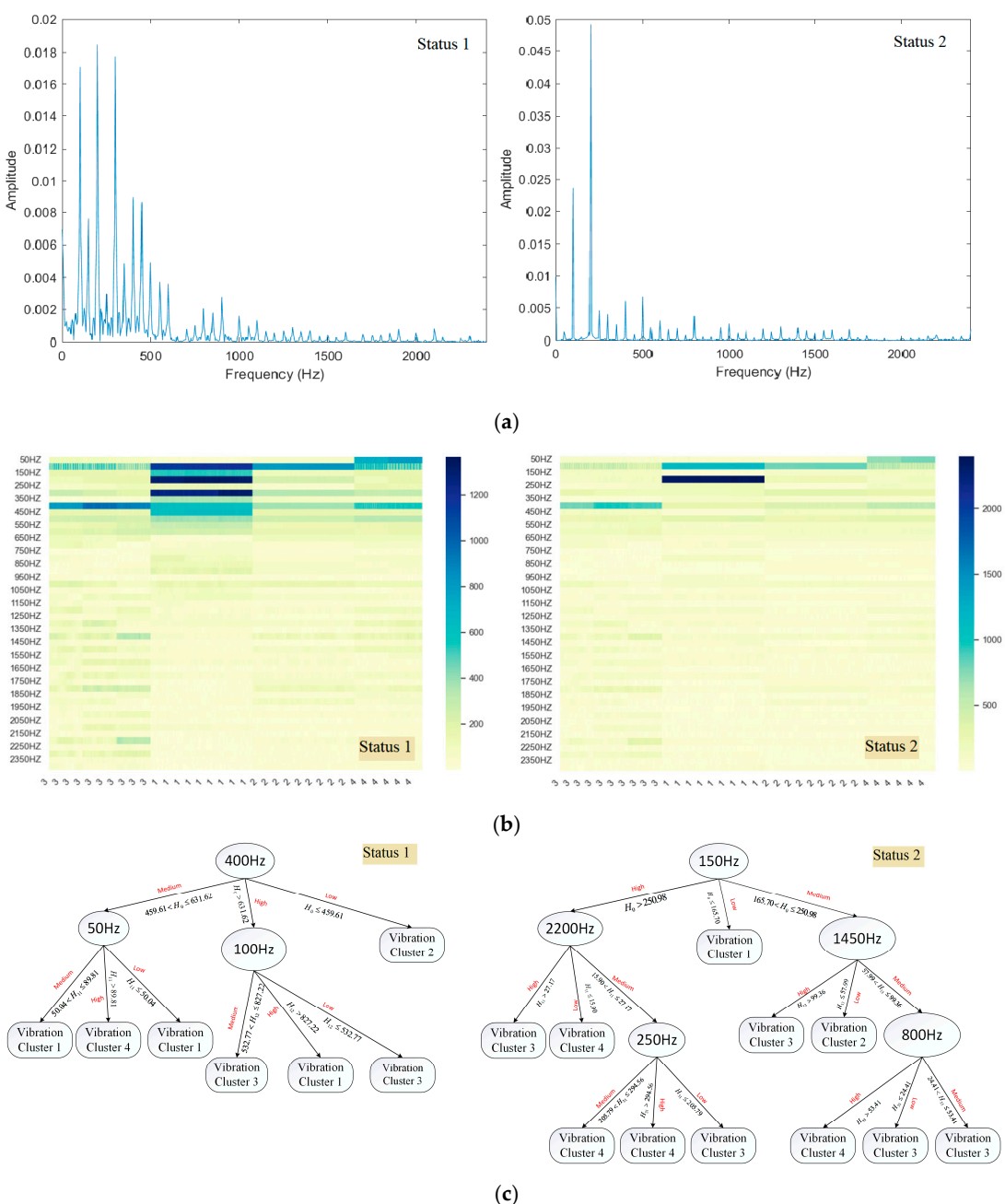

**Figure 8.** Vibration characteristics of transformer in two unknown states. (**a**) Spectrogram of transformers. (**b**) Transformer clustering heat map. (**c**) Decision tree model for a state of a transformer.

Figure 8c shows that the range of $H_0, H_{11}, H_{12}, H_2$ changes to varying degrees in both unknown states of the transformer, especially in the light load condition where the spectral components are beyond the normal $H_0 \leq 365.12$ range. Therefore, the light load condition in state 1 cannot be directly classified on the basis of the vibration component characteristics at 400 Hz, but needs to be further determined by the vibration component characteristics at 50 Hz and 100 Hz to distinguish it from the three-phase unbalance and harmonic currents, respectively; the root node of the decision tree constructed in state 2 becomes 150 Hz and the light load condition is directly classified according to the

vibration component characteristics at 150 Hz, which is a very different condition from the normal operating condition.

In summary, the two unknown states of the transformer are judged to be some kind of fault in the light load condition, which needs to be dealt with in a fault alarm. The abnormal vibration of the transformer's internal silicon steel sheet due to loose screws will cause a relative increase in magnetostrictive strain values, and the vibration component in the 100 Hz to 500 Hz frequency range generated by the core may increase accordingly. In the event of a loose winding fault in the transformer winding, the vibration component at 100 Hz will increase relatively due to the dynamic Lorentz force on the winding and will also increase at 200 Hz, 300 Hz, and other frequencies due to the nonlinear mechanics of the pad. Therefore, in combination with the change in vibration characteristics and the analysis of the vibration mechanism, it is assumed that the transformer generates a mechanical fault in both states.

The magnetic flux density and the magnetostriction of the transformer silicon steel sheet are strongly correlated; the higher the flux density, the stronger the magnetostriction. The transformer flux density diagram for both states is shown in Figure 9a. The maximum value of the main flux density of the transformer in both modes is more than 2.0 T, which is outside of the normal working range of the transformer. The main flux density of this transformer is approximately 1.7 T when operating under normal conditions. The transformer's stress distribution is depicted in Figure 9b, and the transformer winding exhibits varying degrees of deformation in both modes. With the study in Figure 9a,b combined, it is clear that the transformer's aberrant vibration characteristics in both modes are caused by internal mechanical flaws.

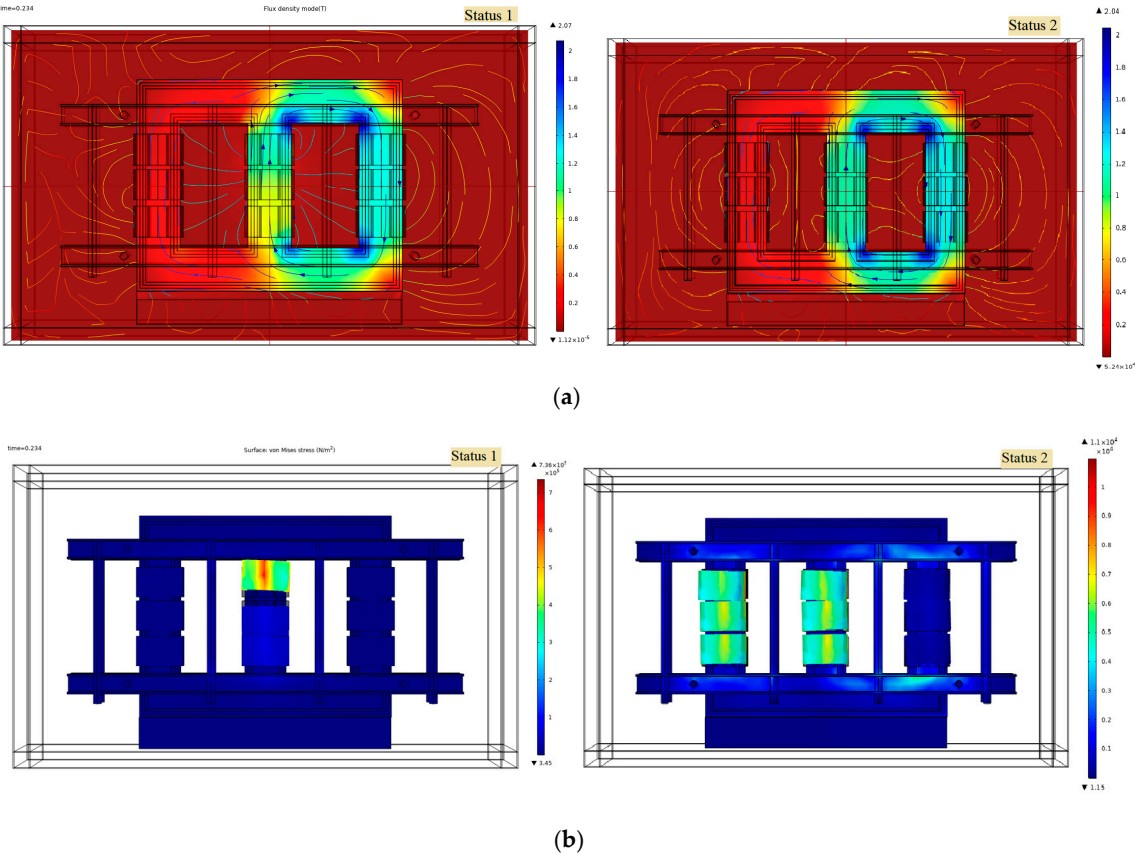

**Figure 9.** Multiphysical field simulation diagram for two states of the transformer. (**a**) Magnetic flux density diagram. (**b**) Stress distribution in a transformer fault condition.

The three fault warning models, SVM, KNN and K-means, were trained with empirical parameters and the test results were compared with the SC−ID3 model and the comparison

results are shown in Figure 10. According to Figure 10, it can be seen that the SC−ID3 model has the highest recognition accuracy among these four models, and it can also show good recognition accuracy under the two working conditions of light load and heavy load where the parameters are more similar.

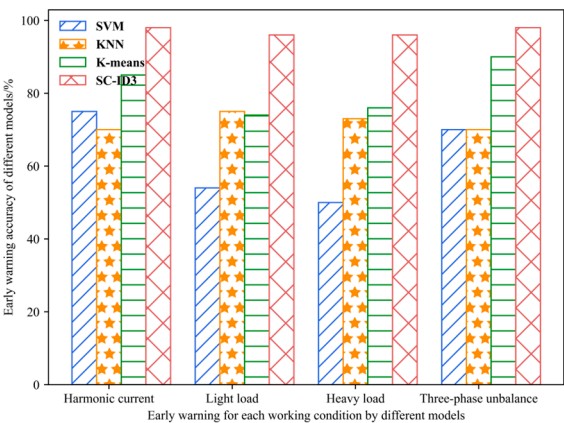

**Figure 10.** Comparison of multiple model recognition results.

### 6. Conclusions

For the transformer vibration signals under different operating conditions, this paper has carried out a feature clustering process based on the spectral clustering algorithm to obtain four vibration clusters combined with data sets labelled with operating conditions. The vibration characteristics under different operating conditions are analyzed by clustering heat maps and building a decision tree model to determine the main information sources to distinguish the operating conditions. The following conclusions were drawn from the vibration characterization system set up in this paper:

(1) The actual transformer vibration signals collected usually do not contain a priori information on obtaining the operating state, and the spectral clustering algorithm can achieve objective and accurate classification of each transformer operating condition data in the vibration signal spectrum data set.

(2) The decision tree model of transformer vibration characteristics can be used for the early warning of faults under different operating conditions of transformers. At this point, however, it is important to build a multiphysical field simulation model of electric-magnetic forces to test how well the fault sources have been identified.

(3) Based on the vibration analysis system set up in this paper, the vibration state parameters of power transformers can be analyzed online to determine how the transformers are working and what their internal health status is. This gives us a new way to monitor the condition of power transformers and find problems before they happen.

**Author Contributions:** Conceptualization, L.R.; Writing—original draft, J.C.; Project administration, H.L., J.L. and L.S.; Funding acquisition, L.Z. All authors have read and agreed to the published version of the manuscript.

**Funding:** This research was funded by the National Natural Science Foundation of China, grant number 52277016.

**Data Availability Statement:** Experimentally collected data is not applicable due to data privacy issues.

**Conflicts of Interest:** The authors declare no conflict of interest.

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
