# Peer review of "Transformer Fault Warning Based on Spectral Clustering and Decision Tree"

_electronics, doi:10.3390/electronics12020265_

Round 1
Reviewer 1 Report
1. one most essential parameters of this research is vibration data. however, there is no explanation about the data set mainly how vibration signals are obtained.
2. In figure 3, the author provides four different curves for four other conditions. the author should justify why and how they differ based on conditions.
3. how the decision tree is applied? the author should explain it briefly with parameters in section 4.
4. In section 5, "The spectrum of each frequency acoustic wave in the range of 50Hz–2400 Hz was extracted as the vibration feature vector, and the FFT was calculated to obtain a total of 440 sets of sliced data sets." how this frequency range is selected and why? the FFT was calculated to obtain a total of 440 sets of sliced data sets. explain why?
5. The proposed method is not compared with the related state of arts.
6. This method has to verify using the standard data set.
Reviewer 2 Report
1. The introduction must contain a proper literature survey and research gap explanation.
2. The authors must add some other results and their discussions.3.
3. More explanation and experimental validation results are required.
4. The novelty is poorly explained.
5. The methodology needs to be explained in detail.
6. The conclusion must contain quantitative analysis rater than qualitative analysis.
Reviewer 3 Report
1. Authors should polish their language usage in this paper. Although there are not a few grammar errors, it is not appropriate for English usage.
2. Please improve the figures in your paper. It is not clear to see these figures, such as Fig. 7 and Fig.8 (a) and (b).
3. Can you explain how all the values in Table 1 can be obtained? they are pre-set or been calculated by some algorithms?
4. Gain=1.392, how to calculate it?
5. Why did you just choose the frequency in the range of 50HZ to 2400HZ?
6. The most challenging is that you did not compare the proposed methods to any other existing method. It is not persuadable to validate your methods. if possible, please add some comparisons.
Round 2
Reviewer 1 Report
Thanks for your response. You need to address more on my comments 5 and 6.
1. 'Comments 5: The proposed method is not compared with the related state of arts.' For this comment, please use related research to compare your research.
2. 'Comments 6: This method has to verify using the standard data set.' if you do not find the other data set then justify your data set i.e. explain how your data set is standard.
Reviewer 2 Report
Accept
